# Premature STEMI in Men and Women: Current Clinical Features and Improvements in Management and Prognosis

**DOI:** 10.3390/jcm10061314

**Published:** 2021-03-23

**Authors:** Rebeca Lorca, Isaac Pascual, Andrea Aparicio, Alejandro Junco-Vicente, Rut Alvarez-Velasco, Noemi Barja, Luis Roces, Alfonso Suárez-Cuervo, Rocio Diaz, Cesar Moris, Daniel Hernandez-Vaquero, Pablo Avanzas

**Affiliations:** 1Heart Area, Hospital Universitario Central de Asturias, 33011 Oviedo, Spain; lorcarebeca@gmail.com (R.L.); apariciogavilanes@gmail.com (A.A.); ajuncovicente@gmail.com (A.J.-V.); rutalvarez3@gmail.com (R.A.-V.); noeminbg@gmail.com (N.B.); alfonsosuarezcuervi@yahoo.es (A.S.-C.); diazmendezro@gmail.com (R.D.); cesarmoris@gmail.com (C.M.); dhvaquero@gmail.com (D.H.-V.); avanzas@secardiologia.es (P.A.); 2Instituto de Investigación Sanitaria del Princpado de Asturias, ISPA, 33011 Oviedo, Spain; 3Anestesiología, Reanimación y Terapéutica del Dolor, Completo Asistencial Universitario de Salamanca, 37007 Salamanca, Spain; luisrocessoto@gmail.com

**Keywords:** coronary artery disease (CAD), ST-segment–elevation myocardial infarction (STEMI), premature STEMI

## Abstract

**Background:** Coronary artery disease (CAD) is the most frequent cause of ST-segment elevation myocardial infarction (STEMI). Etiopathogenic and prognostic characteristics in young patients may differ from older patients and young women may present worse outcomes than men. We aimed to evaluate the clinical characteristics and prognosis of men and women with premature STEMI. Methods: A total 1404 consecutive patients were referred to our institution for emergency cardiac catheterization due to STEMI suspicion (1 January 2014–31 December 2018). Patients with confirmed premature (<55 years old in men and <60 in women) STEMI (366 patients, 83% men and 17% women) were included (359 atherothrombotic and 7 spontaneous coronary artery dissection (SCAD)). Results: Premature STEMI patients had a high prevalence of classical cardiovascular risk factors. Mean follow-up was 4.1 years (±1.75 SD). Mortality rates, re-hospitalization, and hospital stay showed no significant differences between sexes. More than 10% of women with premature STEMI suffered SCAD. There were no significant differences between sexes, neither among cholesterol levels nor in hypolipemiant therapy. The global survival rates were similar to that expected in the general population of the same sex and age in our region with a significantly higher excess of mortality at 6 years among men compared with the general population. Conclusion: Our results showed a high incidence of cardiovascular risk factors, a high prevalence of SCAD among young women, and a generally good prognosis after standardized treatment. During follow-up, 23% suffered a major cardiovascular event (MACE), without significant differences between sexes and observed survival at 1, 3, and 6 years of follow-up was 96.57% (95% CI 94.04–98.04), 95.64% (95% CI 92.87–97.35), and 94.5% (95% CI 91.12–97.66). An extra effort to prevent/delay STEMI should be invested focusing on smoking avoidance and optimal hypolipemiant treatment both in primary and secondary prevention.

## 1. Introduction

Over the past decades, due to primary percutaneous coronary intervention (PCI) programs, modern antithrombotic therapy, and secondary prevention measures, ST-segment elevation myocardial infarction (STEMI) survival rates have significantly improved [1,2]. However, coronary artery disease (CAD) is still considered the most frequent cause of death worldwide and has a huge impact on the health of our population, mainly due to premature death [3,4]. Years of life lost due to premature death accounted for more than 90% of the total disability-adjusted life years (DALYs) due to ischemic heart diseases [4].

Investing in primary prevention for CAD is of utmost importance even in the young population with an incidence of coronary events of 1% per year in men and 0.4% in women between the ages of 30 and 54 years [5]. In DALYs distribution by age, 6% of the total was in persons younger than 45 years and 28% in those aged 45 to 65 years [4]. It has been suggested that etiopathogenic and prognostic characteristics of acute myocardial infarction (AMI) in young patients may differ from those in older patients [6,7]. However, smoking and dyslipidemia have been reported as the most important risk factors of this population [6,8].

Some studies have highlighted that women with STEMI present worse short- and long-term outcomes than men [9,10,11,12,13,14]. In addition, young women, in particular, have worse short-term and long-term outcomes than men and that women continue to receive less-aggressive invasive and pharmacological therapy than men [15].

In this study, we evaluated the clinical characteristics and prognosis of a cohort of patients with premature STEMI and possible differences in regard to the sexes. We aimed to compare the survival rates of men and women with premature STEMI treated with primary PCI with the rates of the general population matching in age, sex, and geographic region. Further, the cholesterol profile and pharmacological therapy in primary and secondary prevention were evaluated in this cohort.

## 2. Materials and Methods

### 2.1. Study Population

We retrospectively identified all consecutive patients with STEMI suspicion referred to our institution for emergency cardiac catheterization from 1 January 2014 to 31 December 2018.

The diagnosis of STEMI was established according to the Fourth Universal Definition of myocardial infarction [16]. The differential diagnosis included other pathologies that can cause ST-segment elevations without significant coronary artery disease, such as myocarditis, pericarditis, stress cardiomyopathy, etc. [16,17].

In this study, only patients with premature STEMI confirmed by the coronary angiogram were included. Premature CAD was considered according to the Dutch Lipid Clinic Network (DLCN) criteria definition for men <55 and women <60 years [18]. Patients without significant coronary artery disease or with major epicardial coronary arteries, either angiographically normal or with documented non-significant disease (<50% stenosis) per coronary angiogram [19], were excluded. Patients with a final diagnosis of non-ST-elevation myocardial infarction (NSTEMI), unstable angina, or other cardiac or extracardiac conditions were also excluded.

From 1 January 2014 to 31 December 2018, 1404 consecutive patients were referred to our institution for emergency cardiac catheterization because of STEMI suspicion. According to the inclusion criteria, 366 consecutive patients with a confirmed diagnosis of premature atherothrombotic STEMI (359 patients) or spontaneous coronary artery dissection (SCAD, 7 patients) were identified by coronary angiography (Figure 1).

All patients were managed by the attending physician, according to current practice guidelines [1].

### 2.2. Data Collection

For this retrospective study, baseline characteristics, in-hospital data, and treatments at hospital discharge were collected from 2014 to date in a prospectively collected database.

Classical cardiovascular risk factors like high blood pressure, tobacco consumption, diabetes, dyslipidemia, and body mass index (BMI) were collected. The patient’s family history of premature cardiovascular disease or hypercholesterolemia in their first-degree relatives, as well as their own cardiovascular history, was investigated. All available LDLc levels were reviewed (prior to STEMI, during admission for STEMI, at discharge, and during follow-up). Patients with STEMI due to a coronary dissection (7 women) were excluded from cholesterol evaluation because of the difference in etiopathogenesis.

Cardiological treatment at discharge was reviewed (diuretics, antiplatelet, anticoagulant, antihypertensive, antiarrhythmic, and hypolipemiant therapy). All patients were treated at discharge following optimal medical treatment according to the European Society of Cardiology Guidelines [1]. Hypolipemiant drugs received prior to STEMI, at discharge, and during follow-up were also reviewed.

All patients were admitted to the same reference center for emergency cardiac catheterization because of STEMI. Most patients were followed-up in this center after discharge, but some of them were followed-up locally at smaller hospitals connected with the reference center via the intranet (so that all medical records can be consulted from either site). Some follow-up cholesterol levels were determined in primary health-care centers. Once again, the primary health-care system was connected via the intranet. All clinical information and test results were unified in the patient’s history and could be easily consulted from any connected institution.

A major cardiovascular event (MACE) was defined as the composite endpoint of a cardiovascular death and readmission because of an acute coronary syndrome or heart failure. Causes of death were defined according to the Academic Research Consortium consensus guidelines [20]. This study was conducted in accordance with the Declaration of Helsinki and was approved by the local Ethical Committee (Principado de Asturias; registry number 2020/184). As the patient information was collected anonymously, institutional review boards waived the need for individual informed consent.

### 2.3. Statistical Analysis

Statistical analyses were performed with the statistical package STATA version 15.1 (Stata Corp LLC, College Station, TX, USA). Descriptive data for continuous variables are presented as median, interquartile range and as frequencies or percentages for categorical variables. The chi-square test or Fisher exact test was used to compare frequencies, whereas differences in continuous variables were evaluated with either Student’s t-test or Mann–Whitney U test. A *p*-value < 0.05 was considered statistically significant. For the survival analysis, Kaplan–Meier and Cox analyses were performed.

The statistical process comparing the survival rate of patients experiencing premature STEMI with that of the general population of the same sex, age, and region was calculated as reported elsewhere [21,22].

## 3. Results

### 3.1. Baseline Characteristics

According to the inclusion criteria, 366 consecutive patients with a confirmed diagnosis of premature atherothrombotic STEMI (359 patients) or SCAD (seven patients) per coronary angiography were identified (Figure 1). Men <55 years represented 83% (303 male patients) of this cohort and women <60 years represented 17% (63 female patients).

There were 13 patients presenting with cardiac arrest who were referred to our institution for emergency cardiac catheterization in this period and had confirmed STEMI as the cause of the cardiac arrest (3.5% of all premature STEMI). The mortality rate in the follow-up of these patients was higher (33%).

The baseline characteristics of the cohort are presented in Table 1. The median follow-up was 4.04 years (2.7–5.6), mean 4.1 (±1.75 SD). Women with premature STEMI were significantly more diabetic than men. The average hospital stay was similar for both sexes, and slightly shorter among women.

More than 1 in every 10 women with premature STEMI suffered a SCAD. Out of those seven women, one was diabetic and two were active smokers. Nearly half of them (43%) presented a previous history of mixed anxiety–depressive disorders. One of them had a cardiac arrest and died during admission. The rest of them did not have other events during follow-up. Further, six out of seven women were treated conservatively, and an angioplasty was performed in only one dissection. In three women, the left anterior descending artery was the vessel responsible (two in the middle segment; one in the first diagonal, with TIMI flow 1, 3, and both); two had the second obtuse marginal affected (both TIMI 2); another one a posterolateral branch (TIMI 0); and the last one, who had a primary angioplasty performed, had a proximal right coronary artery dissection (TIMI 0).

### 3.2. Follow-Up and Survival Analysis

During follow-up, there were 28 deaths, 23 of them of cardiac origin. The global survival showed no differences between sexes (HR: 0.93, 0.27–3.18). During the first 30 days after admission, there were 16 deaths (13 men and 3 women, HR: 2.02 (0.39–10.43) *p* = 0.402). Most patients (71.4%) died due to STEMI complications. There were two sudden deaths, one due to a cerebrovascular accident and one due to sepsis, and three deaths due to cancer progression.

A total of 62 patients were readmitted, mostly because of STEMI (11, 17.54%), non-STEMI (31.5%), or to perform scheduled revascularization (PCI or CABG, 10 patients, 16.13%). Only three patients (4.84%) had to be readmitted due to heart failure. The remaining seven patients were readmitted for another reason or chest pain without ischemia findings.

In total, 68 patients (18.6%) suffered a MACE. There were no significant differences between sexes (19.4% vs. 15.9% HR; 0.86 (0.42–1.75), *p* = 0.391).

The observed survival at 1, 3, and 6 years of follow-up was 96.57% (95% CI 94.04–98.04), 95.64% (95% CI 92.87–97.35), and 94.5% (95% CI 91.12–97.66), respectively. The expected survival at 1, 3, and 6 years of follow-up was 99.68%, 98.96%, and 97.66%, respectively. Figure 2A shows the observed and expected survival. The excess of mortality at 6 years was 3.21% (95% CI 1.12–6.7). If only those patients who survived 2 months after STEMI (349) were analyzed, observed survival at 1, 3, and 6 years of follow-up was 99.41% (95% CI 97.69–99.81), 98.46% (95% CI 96.32–99.36), and 97.31% (95% CI 94.11–98.78), respectively. The expected survival at 1, 3, and 6 years of follow-up was 99.68%, 98.96%, and 97.66%, respectively. Figure 2B shows the observed and expected survival. The excess of mortality at 6 years was 0.37% (95% CI 1.14–3.64).

For men, observed survival at 1, 3, and 6 years of follow-up was 96.19% (95% CI 93.23–97.87), 95.42% (95% CI 92.23–97.32), and 94.09% (95% CI 90.13–96.49), respectively. The expected survival at 1, 3, and 6 years of follow-up was 99.66%, 98.9%, and 97.55%, respectively. The excess of mortality at 6 years was 3.54% (95% CI 1.08–7.6).

For women, observed survival at 1, 3, and 6 years of follow-up was 98.36% (95% CI 88.93–99.77), 96.72% (95% CI 87.52–99.17), and 96.72% (95% CI 87.52–99.17), respectively. The expected survival at 1, 3, and 6 years of follow-up was 99.77%, 99.25%, and 98.28%, respectively. The excess of mortality at 6 years was 1.58% (0.91–10.94).

### 3.3. Cholesterol Profile Management

The distribution of LDLc-available levels prior to STEMI, during STEMI admission, and control LDL levels after STEMI are shown in Table 2. Data were unavailable for those patients without any prior medical visits.

The LDL levels prior to STEMI were available for 205 patients (56% of the patients with premature atherothrombotic STEMI). The LDLc levels evolution of these patients is shown in Figure 3. Among patients with LDLc < 116 prior to STEMI (40 patients), 20% were already under hypolipemiant therapy (Figure 3). At STEMI admission, most of them (91.9%) had LDLc < 116, and only three (8.1%) had LDLc levels between 116–155.

There were 56 patients with LDLc levels between 116 and 155 prior to STEMI who were supposed to have received lifestyle change advice plus statin therapy if insufficient. At STEMI admission, only 12.5% of them (seven patients) were under hypolipemiant therapy. Most of them (58.9%) suffered a myocardial infarction with LDLc levels < 116, whereas 34% presented similar values. In 7.1% of patients, LDLc levels had worsened (155–190) at STEMI admission.

Among patients with LDL levels between 155 and 190 (81 patients), only 20% of them were under treatment. At STEMI admission, 48.8% had LDL levels < 116; 33.75% between 116 and 155; in 16.25%, the LDL remained 155–190; and it was over 190 in one patient.

Only 57.14% of patients who presented LDL levels > 190 (28 patients) and were supposed to receive statin therapy actually received said treatment. At admission, 39% of them had LDLc < 116, another 40% had levels between 116–190, and 22% still had levels > 190.

In summary, at STEMI admission, there were 11 patients with LDLc levels > 190 and only 80% were under statin therapy.

Control LDLc levels after STEMI were available for most patients (88.5% of patients with premature atherothrombotic STEMI, Table 2) 6–12 months after discharge (median time to blood sample: 10 months, IQR: 4–16 months). At discharge, most patients (85.8%) were given high-intensity statins. However, 40% of them did not reach LDL levels < 70. Only 23% of them reached the actual goal of LDLc levels < 55. During follow-up, in 20% of the patients, ezetimibe was associated with statin at discharge, whereas 60.75% of patients were discharged with the same statin they were taking prior to admission. Fibrates were only added in six patients (2.8%). In seven patients, the statin was changed by another statin of similar intensity; in 7.5% of the patients, the statin doses were reduced; and in nine patients, it was changed to a lower-intensity statin. An iPCSK9 was initiated in only two patients (1%).

There were no significant differences between sexes neither among cholesterol levels at any time nor in hypolipemiant therapy at STEMI discharge (Table 3).

## 4. Discussion

The accurate management of young patients with CAD is of utmost importance because of their suggested different etiopathogenic and prognostic characteristics compared with older patients [6,7]. Primary PCI protocols in STEMI and secondary prevention strategies have widely demonstrated an improvement in clinical outcomes including survival [1]. In our cohort of premature STEMI, managed according to current clinical guidelines [1], global survival was similar to that expected in the general population of the same sex and age in our region. Therefore, the extensive effort performed to develop and follow clinical guidelines kept showing its worth in clinical practice.

The most prevalent non-modifiable cardiovascular risk factor of our cohort was the antecedent of the family history of premature cardiovascular disease, present in more than one in every four patients with premature STEMI, without significant difference between sexes. However, in accordance with previous studies, MI (myocardial infarction) in young patients is predominantly composed of male smoker patients (75% were active smokers at STEMI admission and nearly 10% were previous smokers). Hypertension is the second cardiovascular risk factor, followed by diabetes and dyslipidemia.

The LDL cholesterol levels prior to STEMI in this population showed that many patients who may have benefited from intensive hypolipemiant therapy, suffered a STEMI with higher LDLc levels than desired (Figure 3). These findings highlight the important role of classical risk factors in the development of premature STEMI. Although young adults are less aware of the cardiovascular risk, they may be the most beneficiated group of primary preventive care strategies. Primary prevention of smoking and dyslipidemia should be more aggressively promoted to decrease the incidence of premature STEMI [6].

Further, women were considered to represent a distinct, high-risk CAD population with a higher prevalence of traditional and non-traditional cardiovascular risk factors [23]. In our cohort of premature STEMI, all cardiovascular risk factors but diabetes were present in the same proportion in both sexes.

Conflicting results have been reported on the possible existence of sex differences in mortality after myocardial infarction. Perhaps the differences between studies could be explained by the difference in sample sizes, the follow-up time variability, the burden of comorbidities, the access to health insurance, and the management of primary angioplasty. Some studies have suggested that young women with STEMI may present worse short- and long-term outcomes and receive less-aggressive invasive and pharmacological therapy than men [9,10,11,12,13,14,15]. Shehab et al. [24] analyzed 15,532 hospitalized patients with STEMI in the Arabian Gulf region and found that younger women (aged ≤ 65 years) had a higher crude adjusted in-hospital and 1 year mortality rates than younger men and were less likely to receive guideline-recommended pharmacotherapy [24]. Another study analyzing 1,260,200 hospitalizations for STEMI in the United States found that younger women with STEMI (19–49 years of age) experienced higher in-hospital mortality that persisted after a multivariable adjustment [25]. Previous studies mainly reported sex differences in short-term mortality, but after adjusting for age, risk factors, and comorbidities these differences disappeared [26,27,28]. Additionally, some studies showed that young women with myocardial infarction presented higher in-hospital mortality, re-hospitalization, and longer length of stay [29,30,31].

However, our data showed otherwise. Mortality rates, re-hospitalization, and hospital stay showed no significant differences between sexes. Moreover, despite the suspicion that women may have had a less-aggressive pharmacological therapy than men, there were no significant differences between sexes in regard to guideline-recommended pharmacotherapy.

Global survival rates were similar to those expected in the general population of the same sex and age in our region and, when analyzed separately, women had no differences from the general population. It is among men (and not women) with STEMI that there is an excess of mortality: at 6 years it was significantly higher than in the general population. Another interesting study analyzing all kinds of myocardial infarction also found that men had a worse long-term prognosis after a first AMI and a higher 7 year mortality than women [32].

There are several possible explanations for the differences found in our cohort. For instance, there seemed to be a different progression of coronary atherosclerosis between genders [33]. Women are thought to present a lower atheroma burden at the coronary tree than men [34]. In fact, women present more often as a single-vessel lesion disease, whereas men show a greater extension of the coronary artery disease [35]. Additionally, these differences may also help to explain the increased risk of reinfarction in men [35].

Further, the mortality rates during hospitalization and early discharge (in the first 2 months) were of utmost importance. In fact, if only those patients who survived 2 months after STEMI were analyzed, the excess of mortality at 6 years was reduced from 3.21% (95% CI 1.12–6.7) to 0.37% (95% CI 1.14–3.64). Therefore, clinicians should pay particular attention to this particular period after STEMI in young patients. Apart from that, it is remarkable that more than 1 in every 10 women with premature STEMI suffered a SCAD. SCAD is known to be an underdiagnosed and important cause of MI especially in young women [36]. An interesting study with a small cohort found that SCAD accounted for 22% of women with STEMI under the age of 45 [37]. In general, and as performed in our cohort (86%), conservative management is preferred to revascularization [36]. In our cohort, classical cardiovascular risk factors were not that predominant in this subpopulation in which nearly half of them (43%) presented a previous history of mixed anxiety–depressive disorders. Adams et al. found that SCAD affects young women with a paucity of cardiovascular risk factors [38], like in our cohort. They considered the major risk factor for SCAD to be a history of anxiety, depression, or neuropsychiatric illness [38].

## 5. Conclusions

Our results showed a high incidence of cardiovascular risk factors, a high prevalence of SCAD among young women, and a good prognosis after treatment following current recommendations. The global survival of patients with premature STEMI who underwent primary PCI was similar to that expected in the general population of the same sex and age in our region. However, excess of mortality at 6 years was found among men being significantly higher than in the general population.

An extra effort to prevent or delay STEMI should be invested focusing on smoking avoidance and optimal hypolipemiant treatment both in primary and secondary prevention.

## 6. Limitations

This was a retrospective analysis with the limitations inherent to an observational single-center study, limited to a single geographical area. Data were collected retrospectively. In a relatively high percentage of patients, as they were healthy individuals without prior medical contact, pre-STEMI LDL levels were unavailable. The results are only applicable to the studied population (premature STEMI patients who underwent primary PCI in a single academic center). In addition, the sample size was relatively small and the statistical power could be low. Caution should be taken when extrapolating these results. We did not have data on the treatment throughout the follow-up period. We did not have objectively reliable data about diet, physical activity, or treatment adherence, which could be important modifying agents.

## Figures and Tables

**Figure 1 jcm-10-01314-f001:**
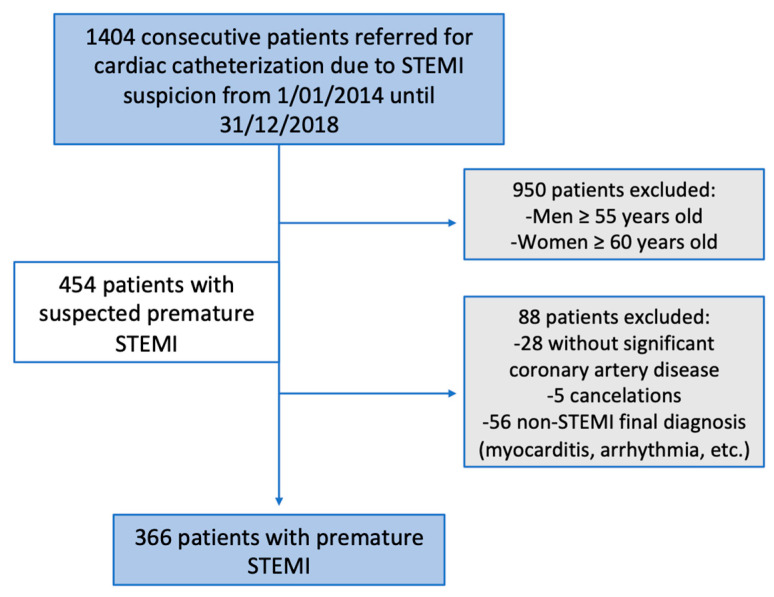
Patient selection. Flowchart showing the steps for the inclusion criteria. STEMI: myocardial infarction with ST-segment-elevation.

**Figure 2 jcm-10-01314-f002:**
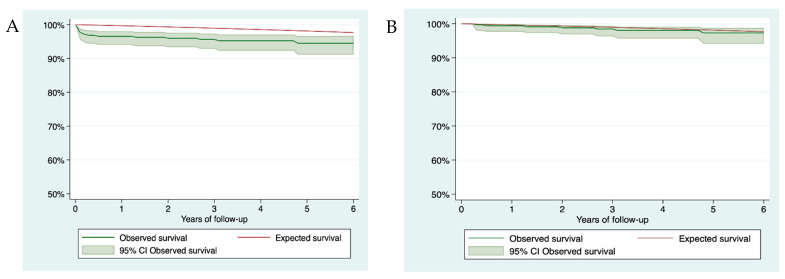
Survival curves (**A**) Global survival curve. (**B**) Survival curve of those alive 2 months after STEMI.

**Figure 3 jcm-10-01314-f003:**
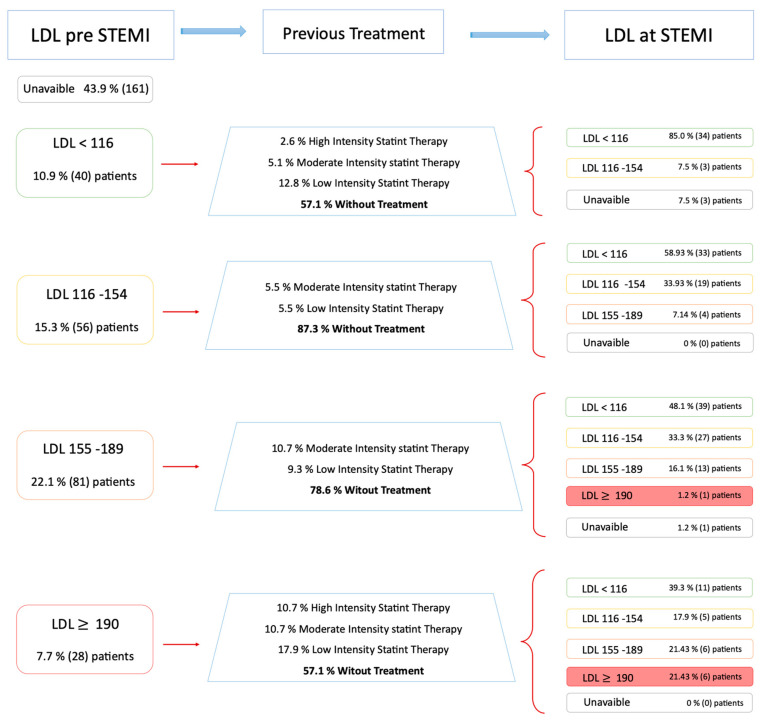
Evolution and management of patients of LDL levels available prior to STEMI.

**Table 1 jcm-10-01314-t001:** Baseline characteristics of the premature STEMI cohort.

	Global (366)	Men (303)	Women (63)	*p* Value
Age (years)	Mean 48.85 (±5.5 SD)	48.3(5.1)	51.6 (6.3)	<0.001
Referred due to cardiac arrest	13 (3.5%)	10 (3.3%)	3 (4.8%)	0.57
BMI	Mean 28.3 (±4.5 SD)	28.45 (4.3)	27.4 (5.2)	0.13
Chronic KIDNEY DISEASE *	2 (0.55%)	2 (0.7%)	0	0.51
Family history of premature cardiovascular disease	106 (29%)	93 (30.7%)	13 (20.6%)	0.11
Cardiovascular risk factors prior to STEMI				
Hypertension	99 (27%)	80 (26.4%)	19 (30.2%)	0.54
Diabetes	35 (9.6%)	22 (7.3%)	13 (20.6%)	0.001
Active smokers Ex-smokers	274 (74.9%) 35 (9.6%)	229 (75.6%) 31 (10.2%)	45 (71.4) 4 (6.35%)	0.08
Dyslipidemia	132 (9.1%)	113 (37.3%)	19 (30.2%)	0.28
Previous LDLc	Median 156 (123–176) Mean 150.09 (±43.15 SD)	156 (122–176)	156 (126–173)	0.66
Previous myocardial infarction	29 (7.9%)	27 (8.9%)	2 (3.2%)	0.19
At admission				
-New Diabetes diagnosis	18 (4.9%)	16 (5.3%)	2 (3.2%)	0.83
-LDLc levels				0.95
	Mean 116.76 (±40.1 SD)	116.70 (39.8)	117.06 (42.1)	
<116	177 (48.4%)	144 (47.5%)	33 (52.4%)	
116–155	101 (27.6%)	92 (30.4%)	9 (14.3%)	
155–190	40 (10.9%)	31 (10.2%)	9 (14.3%)	
≥190	11 (3%)	10 (3.3%)	1 (1.6%)	
Unavailable	37 (10.1%)	26 (8.6%)	11 (17.5%)	
Culprit artery:				0.47
-Left Main	11 (3%)	11 (3.6%)	0	
-LAD	143 (39.1%)	116 (38.3%)	27 (43.85%)	
-LCX	61 (16.7%)	55 (18.15%)	6 (9.5%)	
-RCA	151 (41.3%)	121 (39.9%)	30 (47%)	
Spontaneous coronary artery dissection	7 (1.9%)	0	7 (11.1%)	<0.001
Multivessel Disease	119 (34.4%)	104 (35.7%)	15 (27.3%)	0.23
Average hospital stay	Mean 5.7 (±10.2 SD)	5.76 (±10.4 SD)	5.3 (±9 SD)	0.19
LVEF at discharge	54.55 (± 9.5 SD)	54.50 (±9.25 SD)	54.83 (±10.7 SD)	0.82

LAD: left anterior descending coronary artery. LCX: left circumflex coronary artery. RCA: right coronary artery. LVEF: left ventricular ejection fraction. * Chronic kidney disease is defined as an estimated glomerular filtration rate of less than 60 mL/min/1.73 m^2^.

**Table 2 jcm-10-01314-t002:** LDLc levels prior to STEMI, during STEMI admission, and control LDL levels after STEMI.

	Global	Men	Women	*p* Value
preSTEMI				
<116	40 (10.9%)	34 (11.2%)	6 (17.1%)	0.7979
116–154.9	56 (15.3%)	46 (15.2%)	10 (28.6%)	
155–189.9	81 (22.1%)	68 (22.4%)	13 (20.6%)	
≥190	28 (7.7%)	22 (7.26%)	6 (9.5%)	
Unavailable	161 (43.9%)	133 (43.9%)	28 (44.4%)	
At STEMI				
<116	177 (48.4%)	144 (47.5%)	33 (52.4%)	0.7069
116–154.9	101 (27.6%)	92 (30.4%)	9 (14.3%)	
155–189.9	40 (10.9%)	31 (10.2%)	9 (14.3%)	
≥190	11 (3%)	10 (3.3%)	1 (1.6%)	
Unavailable	37 (10.1%)	26 (8.6%)	11 (17.5%)	
postSTEMI	Global (344)			
<55	82 (23.8%)	70 (24.1%)	12 (22.2%)	0.6912
55–70	63 (18.3%)	52 (17.9%)	11 (20.4%)	
70–116	109 (31.7%)	89 (30.7%)	20 (37%)	
116–190	16 (4.65%)	14 (4.8%)	2 (3.7%)	
Unavailable	74 (21.5%)	65 (22.4%)	9 (16.7%)	

**Table 3 jcm-10-01314-t003:** Statins treatment at discharge from STEMI hospitalization.

Statins at STEMI Discharge	Global (*n* = 344)	Men (*n* = 290)	Women (*n* = 54)	*p* Value
High-intensity statins	295 (85.8%)	246 (84.8%)	49 (90.8%)	0.5471
Medium-intensity statins	15 (4.3%)	13 (4.5%)	3 (3.7%)	
Statins + fibrates or ezetimibe	19 (5.5%)	18 (6.2%)	1 (1.9%)	
Without statins at discharge	2 (0.6%)	2 (0.7%)	0 (0%)	
Unavailable	13 (3.8%)	11 (3.8%)	2 (3.7%)	

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
