# Peer review of "Premature STEMI in Men and Women: Current Clinical Features and Improvements in Management and Prognosis"

_jcm, 2021, doi:10.3390/jcm10061314_

Round 1

Reviewer 1 Report

The authors present the results of a 5 years consecutive evaluation of patients admitted with ST elevation myocardial infarction in a referral centre. The aim of the study is to report on characteristics and outcomes of premature STEMI, defined by the ocurrence in <55 years old in males, and < 60 years old in women. 

The paper is nicely written and the design of the study is correct (restrospective analysis of records). 

I would suggest the authors to remove the part on the comparisons between observed and expected survival as the cohort is to small. If authors wanted to keep the analysis then I would remove the message from abstract and conclusions and might keep it in text. A mortality of 8% (average 4 years FU), 4% (at 30 days) and MACE of 23% in a cohort with a mean age of 48 years old, do not sound something "expected" compared to matched population.

Risk factors on admission are described in detail. Authors also review lipids control after STEMI event and give interesting figures on current lipid control. Very low benefit from powerful drugs (like iPCSK9) despite a significant proportion remain above the recommnedations. 

How about smoking? and hypertension?. Smoking in particular seems to be the number 1 risk factor in this population of premature STEMI. If available, the figure of the patients who quitted smoking habit would be of interest for readers. 

Some comments: 

Abstract need rephrasing. 

first line "Coronary Artery Disease (CAD) is the most frequent cause." please add " of ST-segment-elevation myocardial infarction (STEMI)."

Results: please mention the duration of the follow-up (mean and SD).

Please add years old after <55 in men and <60 in women.

Please change "expect" for "expected in "Global survival rates are similar to that expect in..." please also change "PCI are", for "PCI is"

Please consider changing the line "Global survival rates are similar to that expect in the general population of the same sex and age in our region with a significantly higher excess of mortality at 6 years among men compared to the general population." as this lack of differences might be due to insufficient statistical power (sample size of 359 cases). I would rather give the proportion of patients reaching the main outcome (death or MACE) at 30 days and at 3 or 6 years period. 

Please include in abstract also the number (%) of males and females of the group or premature STEMI.

Introduction: 

please complete phrase in "Investing in primary prevention for CAD is of utmost importance even in the young population, with an incidence of 1%....." please add "of coronary events" after incidence.

Material and methods

please change capital letter in "Differential Diagnosis includes other pathologies that can cause ST-segment elevations..." for "Differencial diagnosis..."

Please change: "All available LDLc levels and hypolipemiant were reviewed.." add "drug" after hypolipemiant

Corrrect etiopathogenesis in the line "due to its different etiopathogenic were excluded to cholesterol evaluation

Please complete statistical methods with Kaplan Meier and Cox analysis details. 

Review format of figure 1 (words appear blurry) 

Results

Please rephrase "Only 60% (13) of young patients with cardiac arrest presented a confirmed STEMI. Mortality rate in follow up of 121 these patients was higher (33%)." it is difficult to understand where these cases come from. 

Please correct: "The rest of them 135 no had other events during follow-up.", change "no had" for " did not have"

Please give information on the angiogram in the 7 cases with SCAD at presentation. Was the artery occluded?, timi flow? "In 6/7 were treated conservatively and angioplasty was performed 136 in only one dissection."

Tables: there is no need to give median and mean. With mean and SD should be enough. please standardize presentation of values with "." decimal point in order to "," decimal comma. One decimal number is enough. 

Please correct "ICP" for "PCI in "...or to perform scheduled revascularization (ICP or CBG, 10 patients, 16.13%)...."

nº MACE: 23 cardiac deaths + 11 STEMI + 31 NSTEMI + 3 HF = 68.
Who are the other 15 patients with MACE? (Did you include scheduled revascularization? If it is the case you should adapt MACE definition. "In total, 83 patients (22,68%) suffered a MACE, without significant differences between genders (23,8% vs 148 17,5 %HR; 0,75 (0,38-1,45), p= 0,391)."

Please add "95% CI" in "...The excess of mortality at 6 years was 1,58% (-165 0,91 - 10,94)."

Please ad "in" before 7.5% and change "doses was" for "doses were" in the line "In 7 patients the statin was changed by another statin of similar intensity, 7.5% of patients their statin doses was reduced and in 9 patients, it was changed by a lower-intensity statin. An iPCSK9 was initiated in only 2 patients (1%)."

Table 3. please change "valor p" "estatinas" "fibratos" "no estatinas al alta" for their english words. correct "unavaible" for "unavailable" 

Discussion

Please correct "performed" "to develop" in the line : "Therefore, the extensive effort perfumed to developed and follow clinical guidelines keep showing it´s worth it in clinical practice."

Change for "benefited" in the line "LDL cholesterol levels prior to STEMI in this population showed that many patient who may have benefit from intensive hypolipemiant therapy,"

Please correct "highlight" in the line "These findings highlights the important role..."

Missing a word "worse" in line "Some studies have suggested that young women with STEMI may present short- and long-term outcomes than men and receive less-aggressive invasive and pharmacological"?

Please "expected" in line: "What is more global survival rates were similar to that expect in the general..."

Please correct "not women" in "It is among men (and no women) with STEMI"

Please correct: "from" and "than more than one" might be more appropriate in the line "Apart form that; it is remarkable more than one..."

Correct "the" and "factors" in the line "They considered that he major risk factor for SCAD are a history of anxiety,.."

Conclussions

Correct "recommendations" "good prognosis after treatment following current recommendation", and expected in "...primary PCI are similar to that expect in the general population of the"

Author Response

The authors present the results of a 5 years consecutive evaluation of patients admitted with ST elevation myocardial infarction in a referral centre. The aim of the study is to report on characteristics and outcomes of premature STEMI, defined by the occurrence in <55 years old in males, and < 60 years old in women.

The paper is nicely written and the design of the study is correct (restrospective analysis of records).

We thank the reviewers for all his/her valuable comments that have helped to improved the quality of the manuscript

I would suggest the authors to remove the part on the comparisons between observed and expected survival as the cohort is too small. If authors wanted to keep the analysis then I would remove the message from abstract and conclusions and might keep it in text.

We thank the reviewer for this valuable comment. As other reviewers have highlighted the importance of these comparison findings, we are unable to remove it from the main text. However, according to the reviewer´s feedback, we have removed the message from abstract and emphasized it in the limitations section.

In addition, sample size is relatively small and statistical power could be low. Caution should be taken when extrapolating these results”.

A mortality of 8% (average 4 years FU), 4% (at 30 days) and MACE of 23% in a cohort with a mean age of 48 years old, do not sound something "expected" compared to matched population.

We thank the reviewer for bringing up this concern. We agree with the reviewer and also believe that a MACE of 18.6% (corrected) is not something “expected” for such a young population. However, we respectfully disagree about the mortality rates. We believe that the 8% described by the reviewer possibly comes from the misunderstanding of dividing the number of total deaths during follow-up / total population. However, for proper evaluation of mortality estimates, the usual methods of k-m or actuarial (like in our case) are the ones preferred. According to that, in our population there was 5.5% mortality (100% - 94.5%) at 6 years, when in the general population of the same age and sex there is 2.34% (100% -97.66%), which represents an excess of mortality of 3.21%.

Risk factors on admission are described in detail. Authors also review lipids control after STEMI event and give interesting figures on current lipid control. Very low benefit from powerful drugs (like iPCSK9) despite a significant proportion remain above the recommnedations.

How about smoking? and hypertension?. Smoking in particular seems to be the number 1 risk factor in this population of premature STEMI. If available, the figure of the patients who quitted smoking habit would be of interest for readers. All patients were advised to quit smoking. However, the number of patients who gave up smoking is unavailable. This has been added to the limitation section.

We thank the reviewer for this valuable comment. We agree with the reviewer in the importance of other cardiovascular risk factors and smoking in particular. Unfortunately reliable data about diet, physical activity or treatment adherence was unavailable in the clinical, as included in the limitations section.

“We do not have objective, reliable data about diet, physical activity or treatment adherence, which could be important modifying agents”

Some comments:

Abstract need rephrasing.

first line "Coronary Artery Disease (CAD) is the most frequent cause." please add " of ST-segment-elevation myocardial infarction (STEMI)."

Corrected

Results: please mention the duration of the follow-up (mean and SD).

Corrected

Please add years old after <55 in men and <60 in women.

Corrected

Please change "expect" for "expected in "Global survival rates are similar to that expect in..." please also change "PCI are", for "PCI is"

Corrected

Please consider changing the line "Global survival rates are similar to that expect in the general population of the same sex and age in our region with a significantly higher excess of mortality at 6 years among men compared to the general population." as this lack of differences might be due to insufficient statistical power (sample size of 359 cases). I would rather give the proportion of patients reaching the main outcome (death or MACE) at 30 days and at 3 or 6 years period.

Corrected

Please include in abstract also the number (%) of males and females of the group or premature STEMI.

Included

Introduction:

please complete phrase in "Investing in primary prevention for CAD is of utmost importance even in the young population, with an incidence of 1%....." please add "of coronary events" after incidence.

Corrected

Material and methods

please change capital letter in "Differential Diagnosis includes other pathologies that can cause ST-segment elevations..." for "Differencial diagnosis…"

Corrected

Please change: "All available LDLc levels and hypolipemiant were reviewed.." add "drug" after hypolipemiant

Corrrect etiopathogenesis in the line "due to its different etiopathogenic were excluded to cholesterol evaluation

Corrected

Please complete statistical methods with Kaplan Meier and Cox analysis details.

Corrected

Review format of figure 1 (words appear blurry)

Corrected

Results

Please rephrase "Only 60% (13) of young patients with cardiac arrest presented a confirmed STEMI. Mortality rate in follow up of 121 these patients was higher (33%)." it is difficult to understand where these cases come from.

Corrected

“Thirteen patients presenting with cardiac arrest were referred to our institution for emergency cardiac catheterization in this period and had a confirmed STEMI as the cause of the cardiac arrest (3.5% of all premature STEMI)”.

Please correct: "The rest of them 135 no had other events during follow-up.", change "no had" for " did not have"

Corrected

Please give information on the angiogram in the 7 cases with SCAD at presentation. Was the artery occluded?, timi flow? "In 6/7 were treated conservatively and angioplasty was performed 136 in only one dissection."

Added

“In 3 women the left anterior descending artery was the responsible vessel (2 middle segment, 1 first diagonal, with TIMI flow 1, 3 and respectively), 2 had the second obtuse marginal affected (both TIMI 2), another one a posterolateral branch (TIMI 0) and the last one, in which primary angioplasty was performed, had a proximal right coronary artery dissection (TIMI 0)”

Tables: there is no need to give median and mean. With mean and SD should be enough. Please standardize presentation of values with "." decimal point in order to "," decimal comma. One decimal number is enough.

Corrected

Please correct "ICP" for "PCI in "...or to perform scheduled revascularization (ICP or CBG, 10 patients, 16.13%).…"

Corrected

nº MACE: 23 cardiac deaths + 11 STEMI + 31 NSTEMI + 3 HF = 68.

Who are the other 15 patients with MACE? (Did you include scheduled revascularization? If it is the case you should adapt MACE definition. "In total, 83 patients (22,68%) suffered a MACE, without significant differences between genders (23,8% vs 148 17,5 %HR; 0,75 (0,38-1,45), p= 0,391)."

We apologize for the mistake. The number of MACE was 68: 23 cardiac deaths (21 STEMI complications and 2 sudden cardiac deaths) + 11 STEMI + 31 NSTEMI + 3 HF.

“In total, 68 patients (18,6%) suffered a MACE, without significant differences between genders (19,4% vs15,9 % HR; 0,86 (0,42-1,75), p= 0,391).

Please add "95% CI" in "...The excess of mortality at 6 years was 1,58% (-165 0,91 – 10,94)."

Corrected

Please ad "in" before 7.5% and change "doses was" for "doses were" in the line "In 7 patients the statin was changed by another statin of similar intensity, 7.5% of patients their statin doses was reduced and in 9 patients, it was changed by a lower-intensity statin. An iPCSK9 was initiated in only 2 patients (1%)."

Corrected

Table 3. please change "valor p" "estatinas" "fibratos" "no estatinas al alta" for their english words. correct "unavaible" for "unavailable"

Corrected

Discussion

Please correct "performed" "to develop" in the line : "Therefore, the extensive effort perfumed to developed and follow clinical guidelines keep showing it´s worth it in clinical practice."

Change for "benefited" in the line "LDL cholesterol levels prior to STEMI in this population showed that many patient who may have benefit from intensive hypolipemiant therapy,"

Corrected

Please correct "highlight" in the line "These findings highlights the important role…"

Corrected

Missing a word "worse" in line "Some studies have suggested that young women with STEMI may present short- and long-term outcomes than men and receive less-aggressive invasive and pharmacological"?

Corrected

Please "expected" in line: "What is more global survival rates were similar to that expect in the general…"

Corrected

Please correct "not women" in "It is among men (and no women) with STEMI"

Corrected

Please correct: "from" and "than more than one" might be more appropriate in the line "Apart form that; it is remarkable more than one…"

Corrected

Correct "the" and "factors" in the line "They considered that he major risk factor for SCAD are a history of anxiety,.."

Corrected

Conclussions

Correct "recommendations" "good prognosis after treatment following current recommendation", and expected in "...primary PCI are similar to that expect in the general population of the"

Corrected

Reviewer 2 Report

Well presented study. Interesting result that authors did not find any significant changes in mortality of younger women with STEMI in contrast to several others contradicting studies also, an excess of mortality at 6 years among men was found to be significantly higher, however authors need to elaborate on possible explanations of their findings, and it's applicability/contribution, to the scientific community.

Amount of information provided for reproducibility of the study seems to be insufficient. For example, a patient's family history of premature CVD was investigated, but nothing is mentioned about using any of family history's parameters for exclusion/inclusion of patients in current study. Also, they mention medical treatment was reviewed but they did not mention for how long, and what were the review parameters? So, they need to elaborate the information in the data collection section.

Author Response

Well presented study. Interesting result that authors did not find any significant changes in mortality of younger women with STEMI in contrast to several others contradicting studies also, an excess of mortality at 6 years among men was found to be significantly higher, however authors need to elaborate on possible explanations of their findings, and it's applicability/contribution, to the scientific community.

We thank the reviewer for this valuable comment and have tried to expand de discussion according to the feed-back provided.

“Conflicting results have been reported on the possible existence of sex differences in mortality after myocardial infarction. Perhaps the differences between studies could be explained due to the difference in sample sizes, the follow-up time variability, the burden of comorbidities, the access to health insurance and also the management of primary angioplasty”.

“Another interesting study (1) analyzing all kind of myocardial infarction, also found that men had a worse long-term prognosis after a first AMI, with a higher 7-year mortality than women.

There are several possible explanations for the difference in our cohort. For instance, there seems to be a different progression of coronary atherosclerosis between genders (2). Women are believe to present a lower atheroma burden at the coronary tree than men (3), presenting more often as a single-vessel lesion disease and men with a greater extension of the coronary artery (4). Besides, these differences may also help to explain the increased risk of reinfarction in men (4)”

  1. García-García C, Molina L, Subirana I, Sala J, Bruguera J, Arós F, et al. Diferencias en función del sexo en las características clínicas, tratamiento y mortalidad a 28 días y 7 años de un primer infarto agudo de miocardio. Estudio RESCATE II. Revista Española de Cardiología. 2014 Jan;67(1):28–35.
  2. Kardys I, Vliegenthart R, Oudkerk M, Hofman A, Witteman JCM. The Female Advantage in Cardiovascular Disease: Do Vascular Beds Contribute Equally? American Journal of Epidemiology. 2007 Jun 14;166(4):403–12.
  3. Nicholls SJ, Wolski K, Sipahi I, Schoenhagen P, Crowe T, Kapadia SR, et al. Rate of progression of coronary atherosclerotic plaque in women. J Am Coll Cardiol. 2007 Apr 10;49(14):1546–51.
  4. Lawesson SS, Stenestrand U, Lagerqvist B, Wallentin L, Swahn E. Gender perspective on risk factors, coronary lesions and long-term outcome in young patients with ST-elevation myocardial infarction. Heart. 2010 Mar;96(6):453–9.

Amount of information provided for reproducibility of the study seems to be insufficient. For example, a patient's family history of premature CVD was investigated, but nothing is mentioned about using any of family history's parameters for exclusion/inclusion of patients in current study.

We thank the reviewer for this comment. We are also concern about the reproducibility. Therefore “Caution should be taken when extrapolating these results” was included in the limitations section.

However, we respectfully disagree about the inclusion criteria. In this study, all consecutive patients with premature (men <55 and women <60 years old) STEMI, confirmed by coronary angiogram, were included. There were no other exclusion/inclusion criteria. Patient's family history of premature CVD was indeed investigated but not considered an exclusion/inclusion of patients

Also, they mention medical treatment was reviewed but they did not mention for how long, and what were the review parameters? So, they need to elaborate the information in the data collection section.

We thank the reviewer for having highlighted this concern. We have expanded the explanation in the data collection section according to the feed-back provided.

"Cardiological treatment at discharge was reviewed (diuretics, antiplatelet, anticoagulant, antihypertensive, antiarrhythmic, and hypolipemiant therapy). All patients were treated at discharge following optimal medical treatment according to European Society of Cardiology Guidelines (1). Hypolipemiant drugs received prior to STEMI, at discharge and during follow-up were also reviewed".

Reviewer 3 Report

Overall good paper.  Interesting contemporary retrospective study of STEMI presentation in younger cohort.

Line 285 of the discussion needs to be revised. 

Author Response

Overall good paper. Interesting contemporary retrospective study of STEMI presentation in younger cohort.

Line 285 of the discussion needs to be revised.

We thank the reviewer for his valuable comment and have modified the line 285 according to the feedback provided

Reviewer 4 Report

I have a few comments and suggestions for the authors:

1.   First of all, it is a pity that there are several careless mistakes (as if the paper as not been reviewed by the authors themselves):

- Line 15: CAD is the most frequent cause… the cause of what?

- Line 135-136 (last sentence of line 135): it should be rather: “The rest of them had no other events…”)

- Line 143: “neoplastic process”

- Line 269: the extensive effort “perfumed” to developed...

- Line 274: hypertension is “the” second…

- Line 306: it is remarkable “that?” more than…

2.   Can you explain in more detail how the data has been collected, especially the follow-up data? I don’t understand when you say that they are “available by intranet”. Does it mean that they have been collected from the medical history records? In this case, don't you think that accuracy is lost and that there may be a lack of rigor/veracity in the results?

3.   Some questions on the results and the discussion.

- In table 1. “Chronic Kidney Disease” is not well defined, what level of glomerular filtration rate did you take into account?

- In the line 121 you say: “Only 60% (13) of young patients with cardiac arrest presented a confirmed STEMI”. I think it is unclear, because the study population is not the patients with cardiac arrest. Moreover in table 1 you mention that 13 patients with premature STEMI were referred due to cardiac arrest.

- In my opinion, you should mention and explain on the discussion the fact that the excess of mortality is mainly due to the mortality in the first 2 months after STEMI (as it is clearly shown in Figure 2).

- I consider it an important limitation that pre-STEMI LDL levels are unavailable for 43.9% of patients. I am not sure that it is correct to analyze LDLc levels evolution based on a small percentage of patients with previous blood tests available, even if I understand it is a limitation inherent to a retrospective study.

Author Response

I have a few comments and suggestions for the authors:

  1. First of all, it is a pity that there are several careless mistakes (as if the paper as not been reviewed by the authors themselves):

- Line 15: CAD is the most frequent cause… the cause of what?          

Corrected

- Line 135-136 (last sentence of line 135): it should be rather: “The rest of them had no other events…”)

Corrected

- Line 143: “neoplastic process”

Corrected

- Line 269: the extensive effort “perfumed” to developed...

Corrected

- Line 274: hypertension is “the” second…

Corrected

- Line 306: it is remarkable “that?” more than…

Corrected

We thank the reviewers for all his/her valuable comments that have helped to improved the quality of the manuscript. We sincerely apologize for these mistakes. We have carefully corrected them all.

  1. Can you explain in more detail how the data has been collected, especially the follow-up data? I don’t understand when you say that they are “available by intranet”. Does it mean that they have been collected from the medical history records? In this case, don't you think that accuracy is lost and that there may be a lack of rigor/veracity in the results?

We thank the reviewer for having highlighted this concern. Therefore, we have expanded the explanation in the data collection section to support the veracity of the results. This study was performed in geographically dispersed region with a main reference center for primary PCI but various smaller centers available for standard outpatient’s follow-ups.

All patients were admitted due to STEMI in the same reference center for emergency cardiac catheterization. Most patients were followed-up after discharge in this center but some of them were followed locally at smaller hospitals connected by intranet with the reference center (so that all medical records can be consulted from either site). Moreover, some follow-up cholesterol levels were determined in primary health-care centers. However, once again, the primary health-care system is connected via intranet. All clinical information and test results are unified in the patient’s history and can be easily consulted from any connected institution”.

  1. Some questions on the results and the discussion.

- In table 1. “Chronic Kidney Disease” is not well defined, what level of glomerular filtration rate did you take into account?

<60, definition.

We thank the reviewers for pointing out the missing definition that we have added to the Table.

Chronic kidney disease (CKD) is defined as an estimated glomerular filtration rate (eGFR) less than 60 ml/min/1.73 m2

- In the line 121 you say: “Only 60% (13) of young patients with cardiac arrest presented a confirmed STEMI”. I think it is unclear, because the study population is not the patients with cardiac arrest. Moreover in table 1 you mention that 13 patients with premature STEMI were referred due to cardiac arrest. According to current guidelines, many cardiac arrest are reffered to our center if the primary cause is suspected to be isquemic, in order to perform a primary PCI. However, many of them did not have an actual STEMI (excluded by coronary angiogram). In this cases other causes such as arrythmias, metabolic, neurological must be investigated.

We apologize for this confusion and have changed the sentence to avoid it. We have removed the percentage of cardiac arrest that were or were not STEMI (as this is an inclusion/exclusion criterion) and decided to directly go on to explain the cardiac arrest due to STEMI.

“Thirteen patients presenting with cardiac arrest were referred to our institution for emergency cardiac catheterization in this period and had a confirmed STEMI as the cause of the cardiac arrest (3.5% of all premature STEMI)”.

- In my opinion, you should mention and explain on the discussion the fact that the excess of mortality is mainly due to the mortality in the first 2 months after STEMI (as it is clearly shown in Figure 2).

We thank the reviewer for raising this point. We agree with the reviewer and have highlighted it in the discussion.

WOn the other hand, the mortality rates during hospitalization and early discharge (in the first 2 months) are of utmost importance. In fact, if only those patients who survived 2 months after STEMI are analyzed, the excess of mortality at 6 years is reduced from a 3,21% (95% CI 1,12 – 6,7) to a 0,37% (95% CI -1,14 - 3,64). Therefore, clinicians should pay particular attention in this particular period after STEMI in the young”.

- I consider it an important limitation that pre-STEMI LDL levels are unavailable for 43.9% of patients. I am not sure that it is correct to analyze LDLc levels evolution based on a small percentage of patients with previous blood tests available, even if I understand it is a limitation inherent to a retrospective study.

We thank the reviewer for this valuable comment and sincerely agree with the reviewer. Unfortunately, many young healthy patients do not have regular medical follow-ups and therefore LDLc are unavailable. As this limitation is impossible to address in a retrospective study, we have added it at the limitations section according to the feedback provided by the reviewer.

“In a relatively high percentage of patients, as they were healthy individuals without prior medical contact, pre-STEMI LDL levels were unavailable”.